# A Spatially Explicit Crop Yield Model to Simulate Agricultural Productivity for Past Societies under Changing Environmental Conditions

**Maarten Van Loo [1],\* and Gert Verstraeten [2]**

[1] Environmental Modelling Unit, Flemish Institute for Technological Research (VITO), 2400 Mol, Belgium
[2] Department of Earth and Environmental Science, KU Leuven, 3001 Heverlee, Belgium; gert.verstraeten@kuleuven.be
\* Correspondence: maarten.vanloo@vito.be

**Abstract:** Most contemporary crop yield models focus on a small time window, operate on a plot location, or do not include the effects of the changing environment, which makes it difficult to use these models to assess the agricultural sustainability for past societies. In this study, adaptions were made to the agronomic AquaCrop model. This adapted model was ran to cover the last 4000 years to simulate the impact of climate and land cover changes, as well as soil dynamics, on the productivity of winter wheat crops for a Mediterranean mountain environment in SW Turkey. AquaCrop has been made spatially explicit, which allows hydrological interactions between different landscape positions, whilst computational time is kept limited by implementing parallelisation schemes on a supercomputer. The adapted model was calibrated and validated using crop and soil information sampled during the 2015 and 2016 harvest periods. Simulated crop yields for the last 4000 years show the strong control of precipitation, while changes in soil thickness following erosion, and to lesser extent re-infiltration of runoff along a slope catena also have a significant impact on crop yield. The latter is especially important in the valleys, where soil and water accumulate. The model results also show that water export to the central valley strongly increased (up to four times) following deforestation and the resulting soil erosion on the hillslopes, turning it into a marsh and rendering it unsuitable for crop cultivation.

**Keywords:** crop yield; soil erosion; modelling; hydrology; spatially explicit; soil thickness; high-performance computing; archaeology

## 1. Introduction

Throughout history, humans and the environment have had a strong interplay in which both greatly impacted the other. The onset of agriculture marked a drastic change in the way humans lived together which resulted in changes in the bio-, geo-, hydro- and atmosphere [1,2]. More specifically, widespread agriculture and the clearance of natural vegetation cover dramatically increased erosion and sedimentation rates [3–6]. Whilst the impact of contemporary soil erosion on soil productivity can be assessed through experimental studies [7], and it has been demonstrated that soil erosion may be a driver of sub-recent land use changes [8], it remains unclear to what extent soil erosion in the more distant past reduced ancient crop productivities to a level that it impacted society. To a large extent, this uncertainty can be related to the fact that a quantification of historic soil erosion remains limited to a few case studies [9].

Furthermore, information on past crop yields is restricted as well. Whilst some proxy datasets give approximations on ancient crop yield, these are of uncertain nature and mostly not applicable in other areas [10–12]. Modelling ancient crop yields could therefore be an alternative solution. However, whilst many models exist to forecast contemporary yields, or future yields under scenarios of climate change, their application on ancient

timescales has received much less attention. Often, such applications are limited in scope, focus on a small time window or operate on a plot location. Furthermore, studies modelling ancient crop yields often do not consider the dynamic nature of the factors controlling crop yield, such as changes in soil quality, soil depth and climate, which makes an integrative approach difficult. The study in [13] models the carrying capacity of the Lower Rhine delta in order to assess whether the Roman army could be supplied with sufficient food. A detailed palaeo-land use reconstruction was used as a basis to assess the productivity of the land, but the analysis was only carried out for two time periods (40–69 CE and 70–140 CE). The paper in [14] model crop yields for the Roman Empire for only a specific period in time (200 AD) using the global hydrological model PCR-GLOBWB [15]. This was done for a large region making use of low resolution HYDE land cover maps and ignoring soil spatial and temporal variability. Low resolution vegetation reconstructions, however, have been proven to have their weaknesses when applied in environmental modelling [16]. The EPIC model [17] has been applied to model crop production in the Copan Valley in the ancient Maya territory [18]. However, whilst the model considers multiple soil types, it does not take into account the spatial interconnectedness of the landscape, nor changes in soil and climatic conditions as it uses present day climate and soil data. Reference [19] combined the SWAT model with an agent-based agricultural model to simulate yields in ancient Mesopotamia under different management scenario's. Although changes in the soil as a result of manuring and irrigation are taken into account, the time span of simulations is limited to only 100 successive years. Reference [20] applied a palaeo-agricultural productivity model developed by [21] in his agent-based modelling of ancient Anasazi population numbers in the Mesa Verde region. Present-day agricultural productivity is related to contemporary Palmer Drought Sensitivity Indices (PDSI). Tree-ring based long-term records of PDSI are then used to calculate ancient productivities at annual timescales. Spatial variability is taken into account using present day soil information; however, this study also lacks the dynamics operating in a landscape on a longer timescale. Reference [22] showed that soil dynamics can be important when modelling crop yield at longer timescales. For a small Mediterranean catchment, they modelled the impact of changes in soil depth following erosion on crop productivity over the last 4000 years. However, only the impact of changes in soil depth on crop yields has been analyzed by [22] whereas other processes were not considered.

Although these studies have their merits, they only take into account a subset of processes that influence crop yield. For processes relevant to understanding the effects of soil erosion on societal sustainability we make a distinction between soil erosion, climate change, hydrology, land management techniques, social processes, soil quality, and pests and diseases. In this study, an attempt was made to incorporate most of the processes, and to take into account the spatial connectivity in the landscape that is inherent when considering the fore mentioned earth surface processes. Here, we present an updated version of the AquaCrop model by making it spatially and temporally explicit thus allowing water availability to be controlled by the position in the landscape. Land cover change, soil erosion, climate change and the dynamics between soil erosion, stoniness and catchment runoff are taken into account.

## 2. Study Area

The ancient city of Sagalassos is situated in the Taurus mountain range, SW Turkey, near Lake Burdur (Figure 1). The territory reached a peak of 1200 km$^2$ during Roman Imperial Times [23], covering a wide range of environments. The landscape is characterised by limestone-dominated mountain ranges and intramontane basins characterised by colluvial, alluvial and lacustrine deposits, whilst mid-slopes are dominated by marl, flysch, ophiolites, mudstone and conglomerate lithologies [24]. The topography ranges between 400 m around Lake Burdur and 2000 m at the higher peaks of the territory. As a result, the climate is highly variable and is characterised by short dry and hot summers and cold, wet winters.

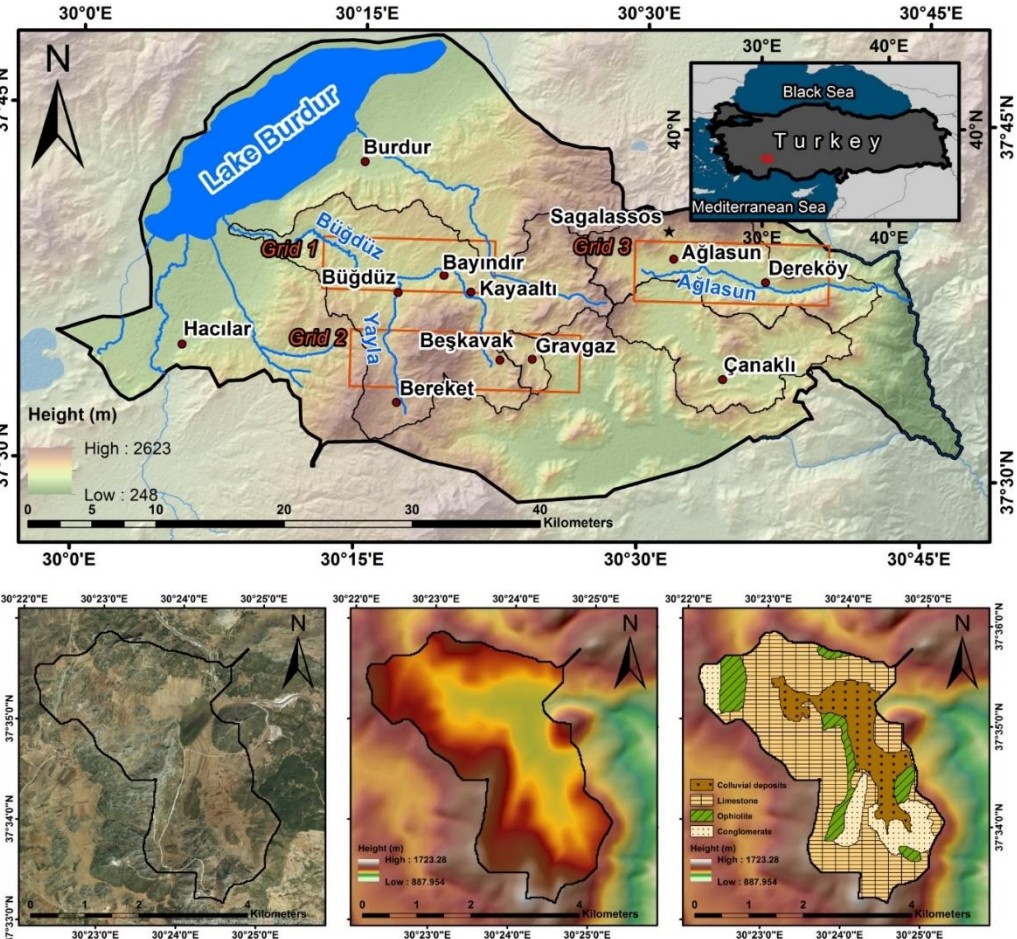

**Figure 1.** Upper panel: Overview of the territory of Sagalassos, main villages, rivers and catchments are indicated. Orange boxes mark the 3 regions where crop yield observations for the calibration of the AquaCrop model were taken during field work (cf. Figure A2). Second row of panels gives an overview of the Gravgaz catchment, where the adapted AquaCrop model was applied on ancient time scales. left panel show the satellite image, the central panel shows the digital elevation model and the right panel the main lithological units.

The fertile valleys are nowadays used for agriculture, with the most dominant plants being *Triticum durum* (wheat), *Cicer arietinum* (chickpea), *Beta vulgaris* (beetroot), *Medicago sativa* (alfalfa) and *Helianthus annuus* (sunflower) [25]. The more remote, less productive fields also often contain *Hordeum vulgare* L. (barley). Most food produced is for local consumption or used as livestock fodder. Archaeobotanical data showed that wheat was one of the important staple crops in ancient times in the territory of Sagalassos [26], hence, it was selected in this study to model changes in crop productivity. Due to the grazing pressure on the higher slopes surrounding the central valleys, shrub vegetation such as *Quercus coccifera* (oak) and *Juniperus oxycedrus* (juniper) is widely found. Vegetation higher up in the mountains (>1200 m) mainly consists of *Pinus nigra* (pine) and some stands of *Cedrus libani* (cedar) and *Abies cilicica* (fir) while lower on the mountain (800–1200 m) *Pinus brutia* (pine) is the dominant species. For a detailed overview of the physiography of the territory of Sagalassos, the reader is invited to consult earlier published work [24,25,27,28].

The Gravgaz catchment within the territory of Sagalassos was chosen to run the spatially explicit erosion and agronomic crop model. Gravgaz is a small endorheic catchment (11.4 km²) (Figure 1) dominated by a wide valley, at present filled with colluvial deposits around a marshy area (1 km²) at an altitude of 1220 m a.s.l. surrounded by limestone hills up to 1590 m a.s.l. During summer, temperatures average around 22.2 °C and precipitation is limited to 13 mm/month, whereas during winter, temperatures drop to 4.1 °C on average,

and precipitation increases to 49 mm/month. At several locations on the lower slopes, conglomerate and ophiolite are present. Several springs border the western side of the marsh, from which several small channels flow towards the northeast border of the marsh where they disappear into karstic outlets. The main karstic outlet in the northeast has been deepened in the last half of the 20th century, which led to the drainage of the Gravgaz basin. Nowadays the lake seasonally swells during winter months, and dries out during summer. Sediment export from the basin is assumed to be insignificant from this endorheic basin [29].

## 3. Modelling Ancient Crop Yields in a Changing Environment

In order to simulate changes in crop yield in the Gravgaz catchment over the last 4000 years in response to changes in climate, land cover and soil quality following soil erosion, the AquaCrop model was used. To apply the AquaCrop model in the study region on ancient time scales, several adaptations were needed. Firstly, adjustments were made to the AquaCrop model to allow it to be spatially connected. Furthermore, runoff and re-infiltration processes were adapted to fit a Mediterranean environment. Lastly, computational times were kept limited by implementing parallelisation schemes on a supercomputer. The open source version of AquaCrop (AquaCropOS version 5.0a [30]) was used as a starting point for model adaptations. The adapted source code can be consulted online at https://github.com/MaartenVL/aquacrop_spatiallyexplicit (available online from the 23 June 2021 onwards). The major changes are briefly described below.

### 3.1. The Adapted Aquacrop Agronomic Model

AquaCrop is a simple water-driven crop yield model that can be used on a variety of soil and climate conditions after local validation [31]. AquaCrop allows the modelling of crop yields with a relatively small amount of input parameters, and as data on past climate, soil and plant functioning are hard to obtain, the AquaCrop model is preferred over more complex models. The model uses canopy ground cover to calculate transpiration, which on its turn is multiplied with the harvest index (HI) to determine crop yield. Soil fertility, water and temperature stresses can reduce crop yield through the crop's transpiration [32]. Several AquaCrop adaptations exist where multiple AquaCrop simulations can be run in batch mode (e.g., AquaGIS, [33], and AquaCropOS, [30]). Although the environmental parameters are sampled from GIS layers such as soil, land use and precipitation maps, the cells on the grid do not communicate with each other.

#### 3.1.1. Runoff and Re-Infiltration

Water availability is often seen as a major restriction to ancient agriculture (e.g., [12,34]). As water availability in a landscape is not simply related to local precipitation but also by runoff from upstream slopes, a 2D runoff scheme has been included in the adapted AquaCrop model. A MATLAB script was made whereby runoff from one cell is distributed to the receiving neighbouring cell using a single-flow algorithm from TopoToolbox [35] (Figure 2). The runoff is simulated in AquaCrop using the curve number approach, which has been adapted here to take into account changes in soil stoniness following soil erosion, thus leading to a dynamic simulation of runoff volumes through time as erosion progresses.

However, not all the runoff that is produced at a hillslope cell will contribute to the runoff at the outlet of a catchment. In semi-arid and patchy Mediterranean environments, re-infiltration of runoff in unsaturated soils may occur leading to a decline in area-specific runoff with increasing slope lengths. Re-infiltration was therefore implemented when routing runoff by considering the upstream contribution of runoff to a grid cell as additional precipitation that can contribute to both infiltration and runoff at that location. However, rates of infiltration and rates of runoff differ greatly [36,37] and to avoid numerical instability when distributing runoff to downstream cells within the model time step, the re-infiltration scheme had to be adjusted. The time that runoff has to pass through a given

cell depends on the flow velocity, which is calculated using Manning's equation based on the local slope derived from a DEM.

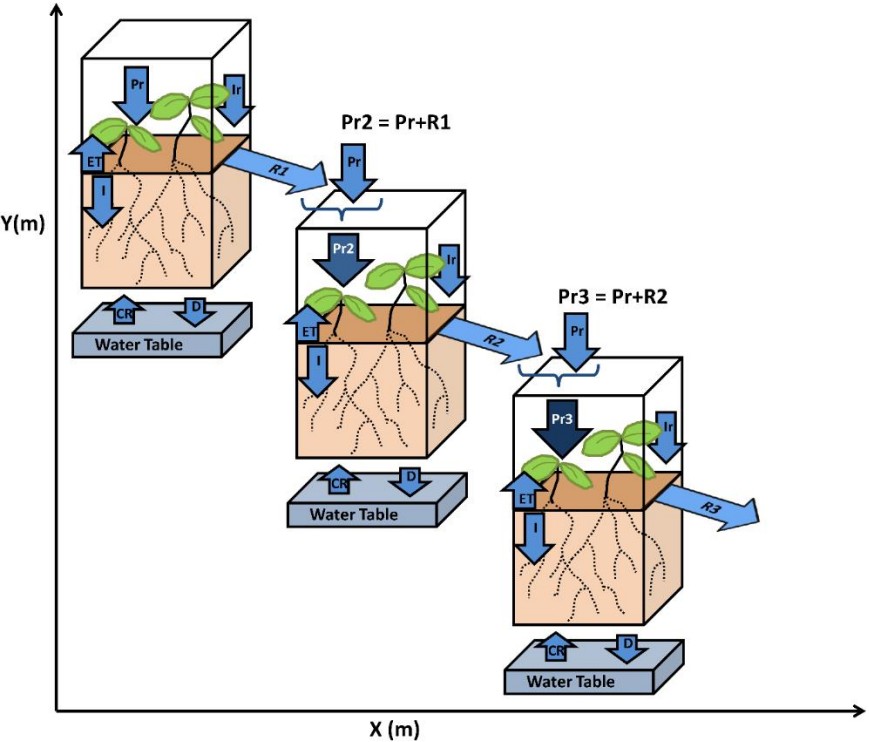

**Figure 2.** Schematic overview of the adapted AquaCrop model used in this study. As runoff (R) accumulates downstream (from the left-most cell to the right-most one), water availability increases. The runoff from the source cell is added to the precipitation (Pr) off the receiving cell. A more detailed description of how much of this additional water can infiltrate is given in the appendices.

Figure 3 gives an overview of the applied re-infiltration scheme. First, AquaCrop calculates runoff (R) and infiltration (Iori) from the precipitation (Ptotal), which is the sum of the precipitation for Cell n (P) and the runoff it receives from an upstream Cell n − 1 (Qinput). The time needed for the flow to pass the cell is then multiplied with the calculated infiltration rate (Iori) to come to the maximum infiltration for that cell. The final infiltration rate (I) is then limited to this maximum infiltration, and the excess Infiltration (Ired) is then added to the originally calculated runoff to come to the corrected and final Runoff (Qcorr), which is passed on to Cell n + 1.

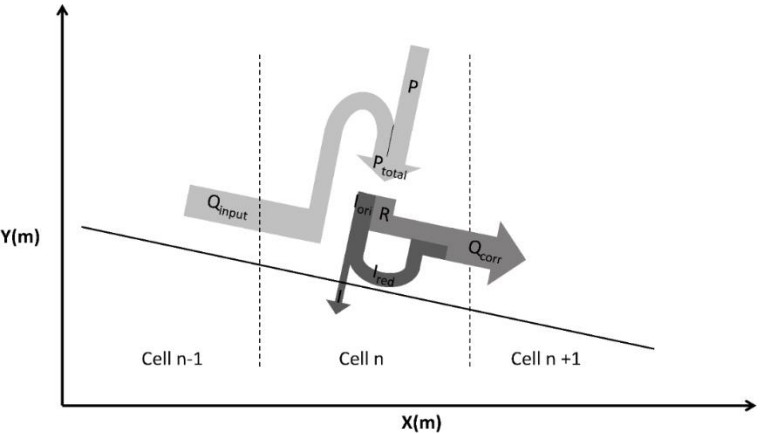

**Figure 3.** Schematic overview of the re-infiltration scheme.

### 3.1.2. Speeding Up Calculations on a Supercomputer

Running AquaCrop in a spatially explicit way and for multiple years requires extensive computational time. Simulations were therefore performed on a Tier-2 high-performance computing cluster (HPC). The HPC cluster allows the performance of multiple calculations of AquaCrop at the same time, a concept also called parallelism. Two different parallel computing concepts were worked out and applied for this study: spatial-parallelism and time-parallelism.

With spatial-parallelism, not all cells in the catchment can be run at the same time, as the results from one cell can depend on the results from another cell. To achieve spatial parallelism on an HPC, the order in which the cells need to be calculated needs to be known a priori. For each receiving cell, the upstream contributing area is calculated with the routing algorithm. AquaCrop calculations are then performed in ascending order of upstream area preventing that a receiving cell is missing information from a source cell. However, cells with the same amount of contributing cells can be run in parallel using a MATLAB parfor loop. This computational scheme is visualised in Figure 4.

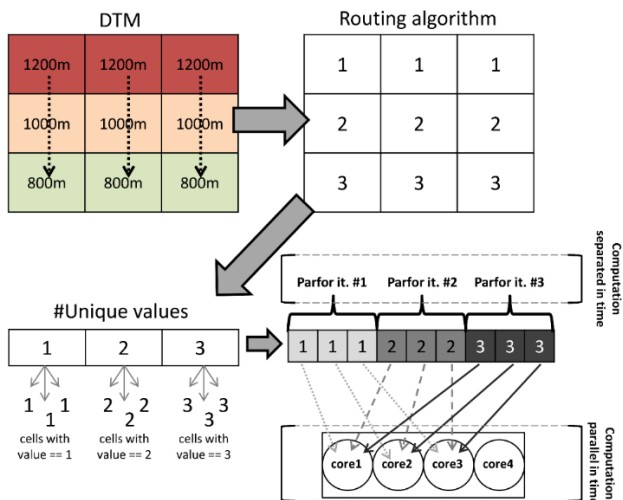

**Figure 4.** Schematic overview of a parfor computational scheme on the HPC-cluster. A routing algorithm calculates for a given cell in the catchment (a receiving cell), the amount of contributing upstream cells (source cells) or upstream area from the DTM. For this example, there are 3 unique values of upstream area (i.e., 1, 2 and 3). A for loop loops through the different unique values of upstream area (in ascending order), hence these calculations are separated in time. Cells with the same upstream area number can be ran in parallel, e.g., all the cells at an altitude of 1200 m or with an upstream area number of 1, can be ran in parallel on 3 different cores.

Due to limitations of the MATLAB parfor loop on the HPC cluster (limited to 20 cores), spatial parallelism is restricted, and time-parallelism can further reduce computation time. Here, several years are run simultaneously on different HPC-cores. The drawback is that the computed status of the soil (e.g., soil water content) at the end of the year cannot be transferred to the next year. However, soils in the study area can be assumed to be relatively dry by the end of the crop season (end of September), which would allow to estimate the soil moisture status of the soil at the start of the next year by assuming dry conditions for the whole catchment. Alternatively, the model itself can be run to find correlations between input variables and SWC at the start of the year. Based on initial model simulations, preformed on a smaller scale but using the parfor-scheme, a soil water content (SWC) prediction model at the grid cell scale was established (Equation (1)):

$$SWC_t = 0.11 + 1.14 * 10^{-5}\,ST_t + 6.38 * 10^{-5}\,P_{t-1} + 2.76 * 10^{-6}\,routing - 0.054 Sand - 6.39 * 10^{-4}\,Slope \qquad (1)$$

with $SWC_t$ (m/m) the SWC at the beginning of year t, $P_{t-1}$ the annual precipitation of the preceding year $t-1$, $ST_t$ the soil thickness (mm) of year t, Sand the sand fraction in

the grid cell, and Slope the local slope (%). Figure 5 shows a relatively good prediction of SWC at the start of a given year. Applying this regression model, a reasonable estimation of SWC at the start of a year can be made without the necessity of running the previous year. The data in the regression equation are available a priori to analysis, which allows it to run the different years in parallel.

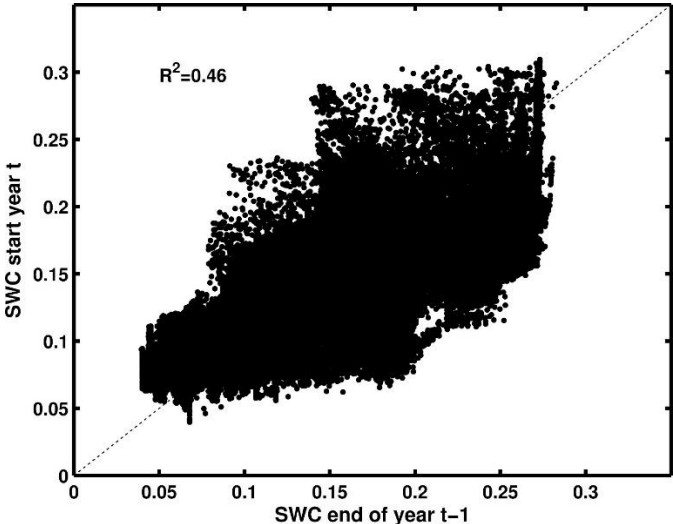

**Figure 5.** Regression equation predicting SWC (m/m) at the beginning of year t based on a priori information (precipitation mm/year) of year t − 1 (Pt − 1), soil thickness (mm) of year t (STt), routing topology (routing), sand fraction (Sand), and the slope (%) (Slope)).

### 3.2. Calibration of the AquaCrop Model in the Environment of Present-Day Sagalassos

The AquaCrop model was ran for winter wheat, and calibrated for local conditions in the territory of Sagalassos. We hereby make the assumption that crop model parameters were the same in the past, and that we can model winter wheat for past conditions without adjusting basic crop parameters to changes in climate and soil, nor taking into account changes in crop properties through natural or anthropogenic selection processes. To calibrate AquaCrop, wheat yields (t/ha/year) were measured in 2015 and 2016 at 38 locations in the study area for a variety of environmental settings (Figure A2). At each site also input data for AquaCrop (slope, stoniness, crop management data) were gathered. More info on the sampling locations and methodology is provided in the appendices. For winter wheat [38], concluded that it would be beneficial to also calibrate the conservative parameters in AquaCrop such as base temperature (Tbase), initial canopy cover (SeedSize), canopy growth rate (CGC), water productivity (WP) and the threshold for cold stress affecting biomass production (GDDmin). A sensitivity analysis of these five parameters was conducted, to check which parameters would benefit most from calibration in the study region. Afterwards, a brute force calibration approach was used to find optimal parameter combinations for the selected parameters. The optimal set of parameters is identified by calculating the Nash and Sutcliffe model efficiency ME [39]. ME is calculated as shown in Equation (2):

$$ME = 1 - \frac{\sum_{i=1}^{n}(O_i - P_i)^2}{\sum_{i=1}^{n}(O_i - P_{mean})^2} \tag{2}$$

Due to the low number of winter wheat crop yield observations, we opted to use a Jackknife calibration approach [40].

### 3.3. Application of the Calibrated Model to Reconstruct Past Crop Yields in Gravgaz

#### 3.3.1. Input Data

To determine the CN of a certain cell, the CN tables from USDA1997 were consulted. Once the cover type is selected, a selection for the Hydrological Soil Group (HSG) needs to be made. [41] assign soils <50 cm to HSG D. Soils between 50 and 100 cm thickness are assigned to HSG A, B or C based on the saturated hydraulic conductivity of the least transmissive layer. Hydraulic conductivity can be calculated when soil texture and stoniness are known. Based on the available soil texture data of samples throughout the territory and a comparison with the lithological map [42], soils on colluvial, limestone and mudstone parent material were classified as loam soils, soils on ophiolite and conglomerate parent material as sandy loam soils. Soils on marl parent material as silty clay loam. By linking soil thickness to stoniness using equations from [43] (Figure 6), and stoniness to hydraulic conductivity, soil thickness will also be dynamically controlling the amount of runoff. The Saxton–Rawls equations for soil water characteristics [44] were consulted to calculate saturated hydraulic conductivity. The computations were done for loam, sandy loam soils and silty clay loam soils, varying the stoniness in the equation. The relation between stoniness and saturated hydraulic conductivity is shown in Figure 6.

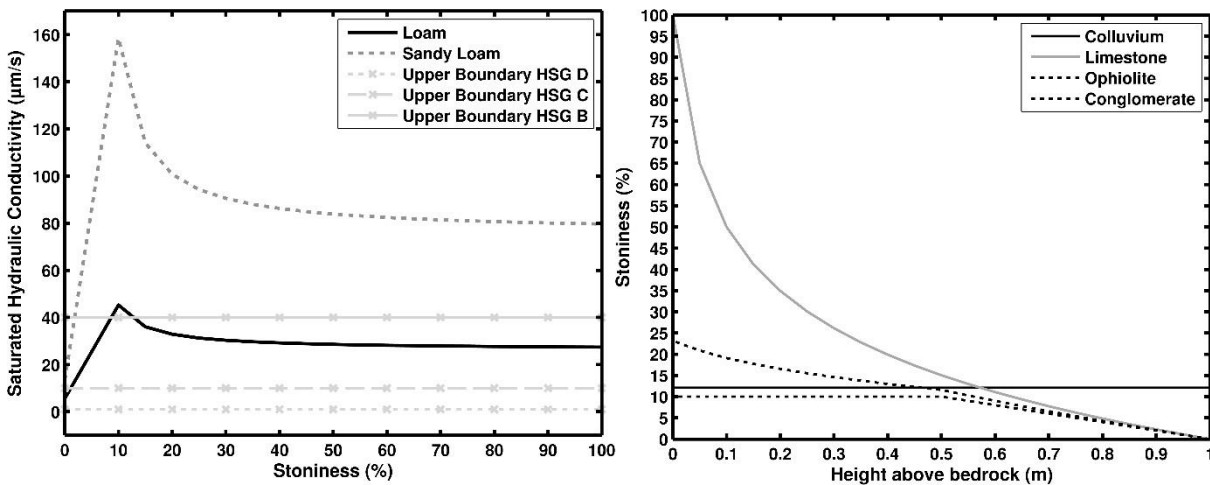

**Figure 6.** Left: Relationship between stoniness (%) and saturated hydraulic conductivity (μS). Based on equations from Saxton and Rawls [44]. HSG's are marked by the horizontal lines. Right: Relationship between height above bedrock (m) and stoniness (%). Based on equations from Dusar et al. [43].

For calculating the flow velocities in the Manning equation, the hydraulic radius for slopes in a Mediterranean mountainous environment were estimated at 5 mm. Manning's roughness coefficient was estimated for both undegraded and the degraded land cover categories based on values from [45–47] (Table 1). For the degraded land cover category, Manning's n was varied depending on the crop development stage. Outside crop development, Manning's r for the degraded land cover category was set to bare soil. Slope was derived from the DTM. Dividing the length of the cell (20 m) by the velocity gives the time needed for runoff to pass to a next cell. We furthermore assume that all rainfall falls during a 1 h rainfall event.

**Table 1.** The Manning's n roughness coefficients used in this study for the degraded and undegraded land cover categories. Outside crop development, the Manning's n values for bare soil are used. During crop development, Manning's n is scaled between the value for bare soil and fully developed wheat, depending on the developmental stage.

| Land Cover Category | Manning's n | Reference |
|---|---|---|
| Undegraded | 0.4 | [47] |
| Degraded (Bare soil) | 0.03 | [45] |
| Degraded (Crop development) | 0.03–0.3 | [46] |

Whilst AquaCrop is essentially a crop growth model, it is here also used to simulate the hydrology of non-agricultural land cover types as this may impact re-infiltration in downstream agricultural grid cells and the water balance of areas with large contribution areas such as valley bottoms. Therefore, AquaCrop was adapted to mimic the effects of trees and shrubs on the water balance. Runoff was adapted through the curve numbers consulted from USDA (1997) for typical native trees in the region (more info in Appendix A). These adaptations are far from complete, but serve as a first attempt to differentiate the hydrology of the undegraded land cover category from the degraded land cover category. Given the absence of hydrogeological mapping of the catchment and given that runoff is included in the adaptations, which will have the greatest impact on the downstream marsh hydrology (expect for the karstic water sources), we argue that for the aims of this study the rather minimalistic model adaptations are justified.

Van Loo et al. [22] simulated the impact of land cover changes and climate on soil erosion and resulting soil thickness. The temporal changes in soil thickness for the Gravgaz catchment were taken from [22] as input data for the AquaCrop model. Soil texture was derived per main lithology, as described in [24]. Changes in land cover and the spatial patterns of degraded (i.e., cropland and grazing areas) and undegraded (i.e., tree cover) land cover through time were also taken from [22] and were based on pollen-based vegetation reconstructions. On degraded land, wheat production was simulated.

Crop parameters were derived from the calibration of the AquaCrop model in the environment of present-day Sagalassos. For the phenological dates, the information gained from the field work campaigns was used: farmers were asked for the dates of planting, emergence, flowering and maturity. The other phenological phases were estimated based on the standard wheat crop file in AquaCrop. Crop yield was modelled for 200 time slices, starting 4000 years ago, with 20 years in between each simulation. The palaeoclimatological timeseries are obtained from a climate reconstruction by [48]. The ECBilt-CLIO-VECODE model was used to simulate monthly rainfall and temperature data over the past 9000 year whereby atmosphere, sea ice, ocean and vegetation are coupled. The coarse 5.6° by 5.6° ECBilt-CLIO-VECODE resolution needed to be downscaled in order to get relevant climate data for the Gravgaz catchment. Details on how this timeseries was downscaled and converted to a daily timescale can be found in the appendices. The resulting precipitation and temperature time series for Gravgaz is shown in Figure 7 in blue and red.

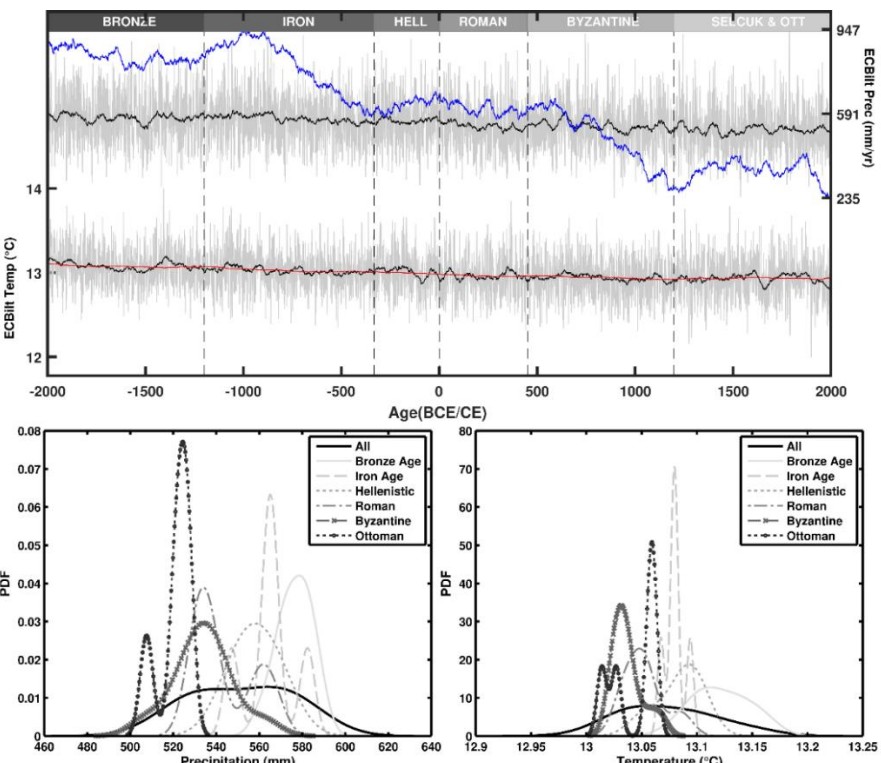

**Figure 7.** (**Top**): Time series of yearly temperature and precipitation from the ECBilt model, with a moving average of 51 years in grey. Lines in blue and red indicate the downscaled precipitation and temperature timeseries for the Gravgaz catchment that were obtained through the WeaGETS stochastic weather generator. (**Bottom**): PDF's of the temperature and precipitation for the different archaeological periods.

### 3.3.2. Modelling Marsh Height and Extent

With water accumulating in the central marsh, there is a potential risk for crop failure due to flooding. In an attempt to quantify this effect, the size that the marsh occupies through time will be modelled based on the runoff entering the central depression. Relations between marsh height, volume and area were retrieved from the DTM (Equations (3) and (4)).

$$\text{flooded marsh area} = 634.5 \times \text{Volume}^{0.31} \tag{3}$$

$$\text{Water height} = 0.0025 \times \text{Volume}^{0.37} \tag{4}$$

Both equations were applied on a daily time step. The volume of the marsh is calculated with Equation (5):

$$\text{Vi} = \text{Vi} - 1 + \text{Run}i + \text{P}i + \text{Spring}i - \text{Evap}i - \text{Karst}i - \text{Infill}i \tag{5}$$

with V the volume of the marsh on the previous day (m$^3$), Run$i$ the total runoff entering the marsh calculated with the adjusted AquaCrop model (m$^3$/day), P$i$ the daily precipitation over the marsh, Spring$i$ the water entering the marsh from adjacent springs (m$^3$/day), Evap$i$ the evaporation from the marsh (m$^3$/day), Karst$i$ the water leaving the marsh through karstic outlets (m$^3$/day) and Infill$i$ the total Infiltration of the marsh (m$^3$/day).

Spring$i$, Karst$i$ and Infill$i$ are unknown and have to be estimated. Values for flow velocity of spring and karstic outlet were estimated at 0.1 m/s, whereas their respective area was estimated at 0.3 m$^2$. Estimations were made based on present day observations. Infiltration rate was estimated at 0.5 mm/h, based on [49].

The equation is solved on a daily time scale. First, runoff to the marsh Runi is added to the initial marsh volume Vi − 1 (volume of the marsh at the end of the previous year). This volume is converted to an area, using Equation (3). The Penman equation was used to calculate evaporation over an open water surface in mm/day, using the generated climate time series as input data [50]. Evapi is then calculated by multiplying the area of the marsh with the Penman calculated evaporation rate. The estimated infiltration rate is multiplied by the size of the marsh to come to Infilli. The karstic outlet is assumed to be located at a height of 0.5 m above the lowest point in the catchment, based on the mapping of [51]. Maximum discharge is only reached at 0.5 m marsh water height. Between a marsh water height of 0 and 0.5 m, discharge is linearly interpolated between 0 and the estimated value. Finally, Equation (5) can be solved, and marsh height and extent can be calculated using Equations (3) and (4). The initial volume of the marsh at the start of simulations is set at 0, as the deepest parts of several corings in the marsh show no peat nor clay but detritic accumulation, pointing to a seasonally dry central valley marsh [29].

## 4. Results and Discussion

### 4.1. AquaCrop Calibration of Present-Day Soil Productivity

Based on the sensitivity analysis, crop parameters Tbase, GDD min, CGC and WP were selected for calibration. These show the highest change in model efficiency per unit change in parameter value. Optimal calibration parameters were found for Tbase = 12 °C, GDDmin = 4 °C day$^{-1}$, CGC = 0.04 (fraction GDD$^{-1}$) and WP = 22 g m$^{-2}$. Table 2 lists the optimal calibration parameters in comparison to the calibration parameters used by [38] for winter wheat and the original parameter values for spring wheat [31]. The optimal set of calibration parameters from the jackknife calibration yields a model validation with a ME of 0.51 (Figure 8).

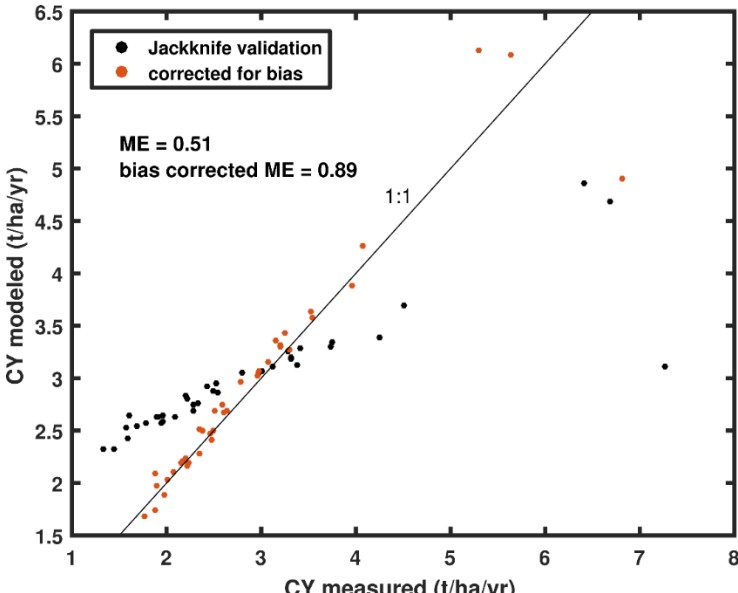

**Figure 8.** Jackknife validation: Measured versus modelled crop yield (ton/ha/year) using the AQ model with optimal parameter values plotted in black. Correction for bias plotted in red. The transformation needed to shift the jackknife validation points to the bias correction is used to correct all crop yield predictions.

**Table 2.** Optimal calibrated crop parameter values. For this study, both the calibration ranges and amount of steps taken during calibration are specified. Tbase: base temperature, GDD: growing degree days, CGC: canopy growth rate, WP: water productivity.

| Crop Parameter | Winter Wheat (This Study) | | | Winter Wheat | Winter Wheat [31] |
|---|---|---|---|---|---|
| | Calibration Ranges (Min–Max) | Amount of Linearly Spaced Calibration Steps | Optimal Value | Optimal Value | Optimal Value |
| Tbase (°C) | 6–12 | 7 | 12 | 2 | 0 |
| GDD min (°C day$^{-1}$) | 0–8 | 9 | 4 | 8 | 14 |
| CGC (fraction GDD$^{-1}$) | 0.001–0.07 | 7 | 0.04 | 0.08 | 0.05 |
| WP (g m$^{-2}$) | 13–22 | 10 | 22 | 18.5 | 15 |

Thus, whilst it is possible to model the general present-day crop yield distribution in the territory reasonably well, plot specific predictions are currently sub-optimal. Figure 8 shows that there is still a systematic bias in the modelled crop productivity values, with low values being overpredicted and higher values underpredicted. Plot specific temperature and precipitation values, initial SWC values, observations of canopy cover throughout the growing season, and a better assessment of soil fertility stress (by using control plots), were not available for this calibration, but would likely help improve plot specific crop yield predictions [52]. In particular the lack of information on local management practices (e.g., weed and pest control, irrigation, use of fertilizers) may be a reason for the systematic bias in the modelled crop productivity. Field with already low yields are often located at more remote locations and these marginal lands may receive less attention by farmers, hence leading to a greater soil fertility stress and model overpredictions. On the other hand, fields that already have higher yields will receive more attention and a better management may lead to higher productivities than modelled.

To overcome the bias in the model predictions, a correction (Equation (6)) is applied to all further crop yield results in the paper. Figure 8 shows the bias-corrected modelled crop yield in red.

$$CY_{corr} = 1.86CY - 2.6 \qquad (6)$$

### 4.2. Modeling Ancient Crop Yield: Spatial and Temporal Patterns of Crop Yield Simulations

Figure 9 shows the modelled time series of catchment average crop yield for Gravgaz using the calibrated AquaCrop model. These crop yield values will mostly be used for relative interpretation throughout time and space, and assessing effects of the drivers of crop yield change. The values presented here should not be interpreted as the actual obtained crop yields during ancient times, given the several assumptions that were made in the modelling process. Crop yield decreases strongly at the start of the Iron Age up until the Hellenistic period, after which it more or less stabilizes. Around the Byzantine period, crop yields increase again. It should be stressed that these are average crop yields for the Gravgaz basin which is the combined effect of changes in crop yield due to changes in soil fertility and climate, as well as changes in land used for agriculture. For instance, during the Roman period, a high fraction of the land was assumed to be taken into cultivation (see the few patches of grey in Figure 10, bottom panels), implying that also a large amount of low productive hillslope areas were used for crop cultivation, lowering the catchment wide crop yield average.

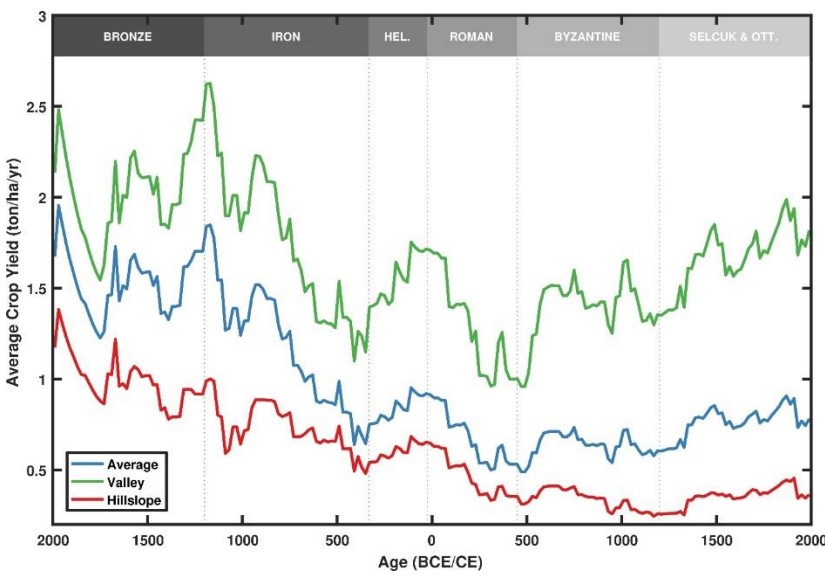

**Figure 9.** Modelled average crop yields. The upper line shows the average yields for the valley areas, the lower one the average yields for the hillslope areas.

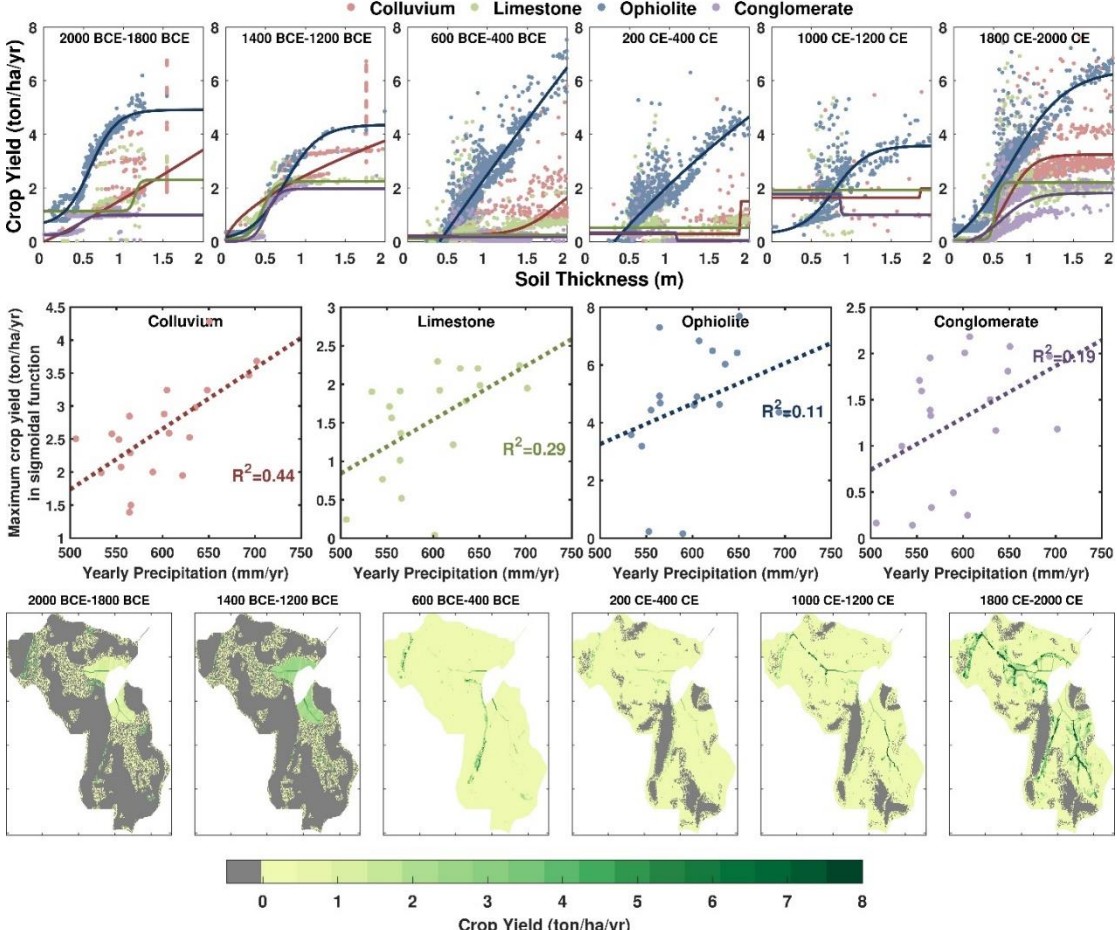

**Figure 10.** Top panels: ST-CY relationships averaged for 200 year time periods, only showing soils up to 2 m. Middle panels: The relationship between yearly precipitation averaged over the 200 year time periods, and the maximum crop yield value in the sigmoidal or logarithmic fits between ST and CY. Precipitation clearly influences the changes in the ST-CY relationship throughout the years. Bottom panels: Average crop yield for 200 year time periods. Cells not under wheat cultivation are shown in grey. Six selected time slices of 200 years are shown.

Crop yield is influenced in a sigmoidal or logarithmic way by soil thickness, as shown in Figure 10 (top panels) for the upper 2 m and at a spatial resolution of 200 years. Not all variance in crop yield can be explained by soil thickness, which can be related to spatial variability in soil lithology and texture, and in particular soil stoniness. In particular soils developed on limestone are more affected by soil degradation than soils on ophiolitic or colluvial material [22]. The relation between crop yield and soil thickness also shows some important temporal changes. Overall, changes in precipitation are the main driver behind the annual relation between crop yield and soil thickness (Figure 10, middle panels). Here as well, this relation is strongest for soils on limestone showing that these soils are more susceptible to climate variability when it comes to sustaining crop yields. The evolution in spatial variability in crop yield for six selected 200-year time slices is shown on Figure 10 as well. Areas covered by trees and shrubs in the simulations are shown in grey. For simplicity, we assume that all the remaining land is under wheat cultivation as this is the only crop we considered in this study, which will likely be an overestimation of the area used for crop cultivation. Valley areas often show up as the areas of highest crop yield, whereas the hillslopes are gradually losing crop yield. Furthermore, areas where water is concentrated are marked by the highest crop yield, and can deliver high yields in periods where the catchment otherwise would have marginal yields (e.g., 600–400 BCE and 1000–1200 CE).

*4.3. Changes in Catchment Hydrology and Land Suitability*

The AquaCrop model not only simulates changes in crop productivity through time but also changes in runoff response following progressive land degradation. Results show that the runoff from hillslopes to the central depression increased up to four times following the soil erosion induced land degradation of the Iron Age (Figure 11). At the end of the Hellenistic period, it increased with 50% again. After this, water export more or less fluctuates around a level of $1 \times 10^6$ m$^3$/year, with peaks during Middle Byzantine times and the end of the Selçuk-Ottoman period. Precipitation strongly controls the amount of water delivered to the central depression; however, this relation has changed fundamentally following the intense land degradation in the Iron Age. For a given amount of rainfall, runoff has increased with 260% when undegraded and degraded conditions are compared. The first peak in land clearance during the Iron Age led to a strong decrease in soil thickness and an increase in soil stoniness [22]. This, in combination with a reduced vegetation cover, will result in higher runoff rates and increased volumes of water delivered to the depression, hence rising the water level. As soil depletion progresses, water export is more susceptible to precipitation changes, as can be seen in the stronger slope of the precipitation—water export relationships as time progresses. Following the changes in hillslope hydrology, the water levels in the depression and the spatial extent of the area covered by water throughout the year, also increases (Figure 12). This also has implications for total productivity values as the fertile flat valley bottoms are no longer accessible. During the Bronze Age, the extension of the marsh causes an average decrease in total valley crop yield of only 27%. However, with the expansion of the marsh during the Roman period, this increases up to 50% (Figure 12, bottom panel).

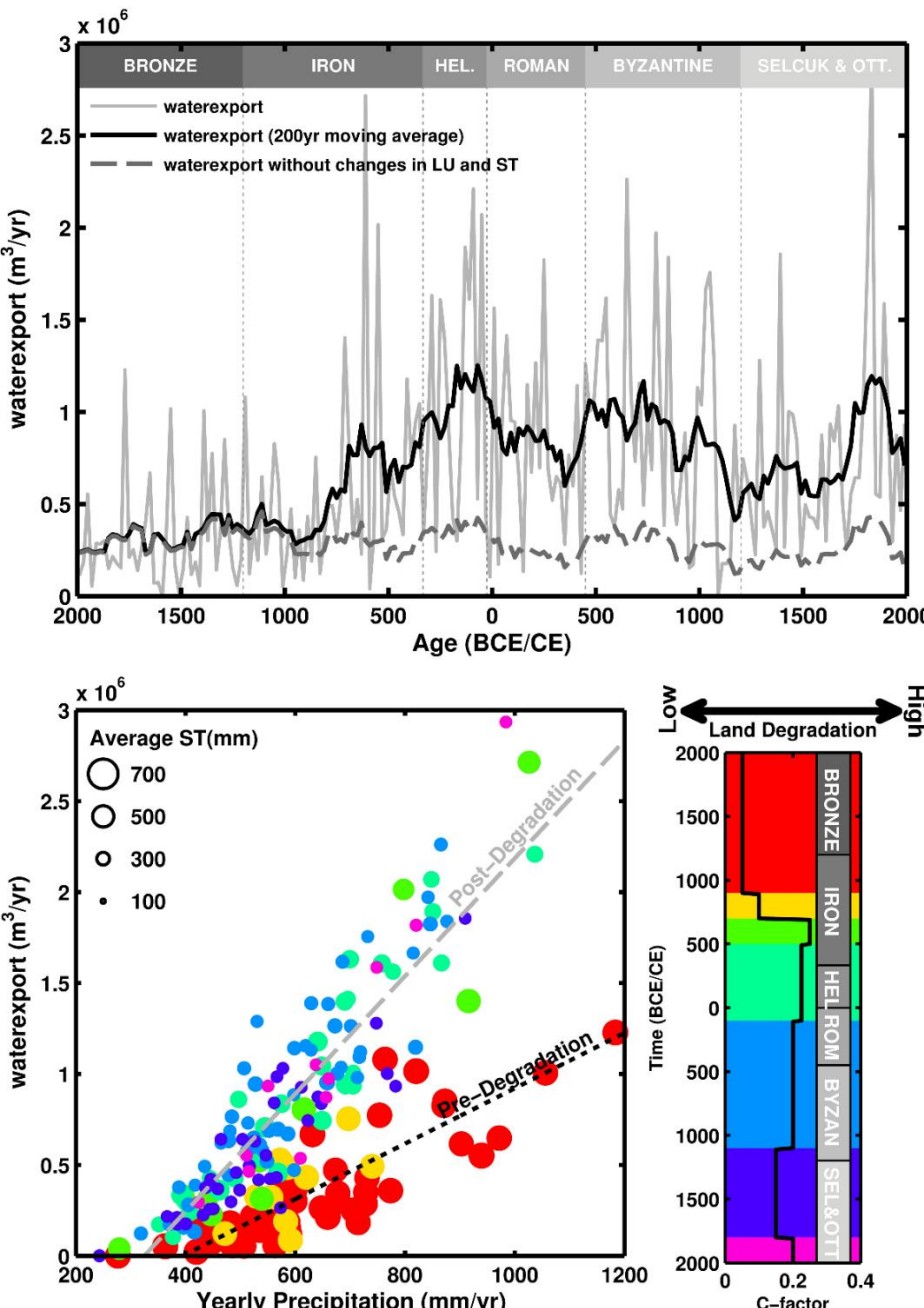

**Figure 11.** Top panel: Modelled water export towards the central valley marsh (m³/year). A moving average of 200 years is plotted in black. Dashed grey line shows the same water export but keeping land use (LU) and soil thickness (ST) constant over time. Around the Middle Iron Age, water export increased to four times its initial value at the beginning of the Bronze Age. If land use and soil thickness are kept constant over time (dashed grey line), this effect is completely absent. Bottom panels: relationship between precipitation (mm/year) and water export (m³/year), land cover (RUSLE C-factor) and average soil thickness on the hillslope areas (mm). The higher the average C-factor in the catchment (bottom-right panel), the higher the fraction degraded land cover.

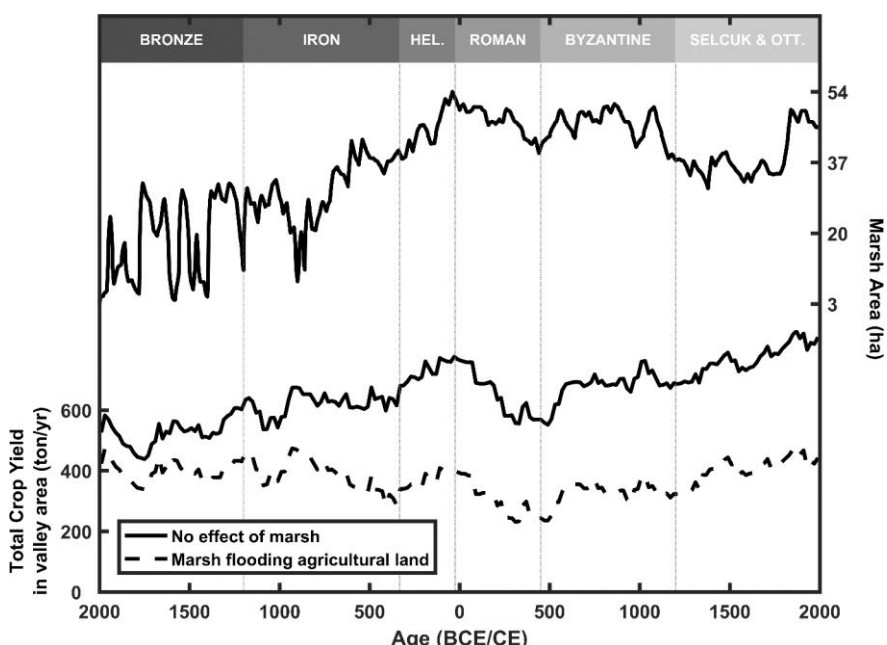

**Figure 12.** Top panel: Size of the marsh (m²) throughout time, or the amount of agricultural land lost by the marsh. Bottom panel: Total crop yield (ton/year) in the valley area, calculated with and without taking into account the loss of agricultural fields flooded by the marsh.

### 4.4. The Value of an Adjusted AquaCrop Model in the Context of Ancient Crop Yield Simulations

The adapted AquaCrop model is able to quantify crop yield on millennial timescales for a small-sized catchment. The application of crop yield models to 2D-landscapes and at longer timescales requires much computing power. Here, we show that the development of a parallel computing scheme significantly speeds up calculations. Table 3 shows the effect of the parallelisation schemes on computing time in hours. The time parallelism scheme using 200 cores sped up calculation times 200 fold. This offers opportunities to model even at large spatial scales, while making less compromises on the complexity or amount of included processes. As knowledge on soil-landscape functioning is ever increasing [9,53–55], it will be of vital importance to not only expand the theoretical bases, but also keep developing platforms that allow to actually enfold these insights. An important adaptation of the AquaCrop model is that it is now fully hydrologically connected. This enables to model the integrated effect of soil and land cover change on crop yields. The results show how important re-infiltration in downstream areas can be in dry periods. In particular for semi-arid environments, but also humid regions with a strong seasonality in rainfall distribution, runoff from upstream areas may contribute significant amounts of water to downslope soils. Many regions in the Mediterranean and the Near East have experienced changes in settlement location throughout history, i.e., from hillslope environments to valley bottoms. It has been suggested that changes in soil quality and crop productivity may lay at the basis of these settlement changes (e.g., [12,56,57]). Our results show that soil erosion is indeed an important driver of changes in crop yield, however, climate variability has the strongest hold on yield variation throughout time (Figure 10). The nature of the relationship nevertheless strongly varies. Whereas precipitation and crop yield show a more or less linear relationship, the relation between crop yield and soil thickness is highly non-linear showing a clear threshold. Consequently, the sensitivity of soil crop productivity to precipitation changes when soil degradation reaches this threshold.

**Table 3.** Computing times (hours) of AquaCrop in different settings.

| | Single AQ Run | Sequential AQ Runs Gravgaz Catchment (28,000 Cells) | Spatial Parallelism Gravgaz Catchment (28,000 Cells) | | | Time Parallelism Gravgaz Catchment (28,000 Cells) |
|---|---|---|---|---|---|---|
| | | | 20 cores | 100 cores | 500 cores | 200 cores |
| 1 year | 0.001 | 38.89 | 1.94 | 0.39 | 0.08 | 38.89 |
| 200 years | 0.278 | 7777.78 | 388.89 | 77.78 | 15.56 | 38.89 |
| 4000 years | 5.556 | 155,555.56 | 7777.78 | 155.56 | 311.11 | 777.78 |

*4.5. Future Work*

Societies have employed a wide variety of land management techniques in order to sustain their agricultural productivity and, hence, their livelihood. Arguably, irrigation, manuring and crop rotation were the most important tools available for ancient farmers to maximize cereal yields in the Mediterranean ([58–60]). Both irrigation and manuring have been detected in the archaeological records (e.g., [12,61]).

The AquaCrop model is capable to model effects of land management, but results of these simulations are not yet presented in this study. With social relationships, we include the processes of exporting and trading crops, as discussed by [14]. Although important to take into account when calculating the carrying capacity of a society, they are less important to the raw calculations of plot scale crop yield, and hence not included in our simulations. Soil quality, although an important factor controlling crop yield, was not taken into account. For the simulations in this paper, we assume a soil that does not suffer from any soil quality stresses. Pest and diseases are known as a stochastic factor that can severely impact crop yields [62]. However, for simplicity, we ignored these effects.

The work of [63] could provide inspiration for future estimates of ancient crop yield where such aspects are taken into account. They provide a framework where carrying capacity is calculated bottom up, using a combination of empirical modelling on the one hand, and a database of high quality historical and archaeological information on the other hand.

When modelling the marsh water height and extent, effects of sedimentation were not taken into account. To model the exact water heights and extent, the initial topography (4000 years ago) of the marsh needs to be known, and ideally be adapted during model simulation as erosion and deposition change the lay of the land. This has been attempted for the WATEM/SEDEM model in the past, but has proven to be a difficult undertaking [64]. The modelling of a dynamic topography unfortunately falls beyond the scope of this study. Furthermore, there is the issue of compaction and burial of the sediments. Sediments will not merely act as volume that reduces the volume of water present in the marsh, but over time can also evolve into a new soil surface, in which water has to infiltrate. Information on the sediment supply to the central valley marsh does exist [22], but the sedimentary chronologies are too limited in number to make a correct assessment of the bathymetry of the marsh.

The assumption of a constant winter wheat variety and managing techniques (sowing densities, tillage techniques, etc.) influences both soil erosion and runoff values. However, the results of the soil erosion modelling validation presented in [22] show that using these assumptions still lead to reasonable results, especially knowing that the main driver in temporal variation of soil erosion was land cover change. With regards to runoff, there are no good data to validate the results. A possible solution would be to match the modelled marsh extent with peat deposits found in the different corings in the valley for which there are sedimentary chronologies. However, this work is largely unfinished due to the difficulties faced in modelling the effect of sedimentation on marsh water height and extent.

## 5. Conclusions

In this study the agronomical model AquaCrop was used for the first time to model crop yields in ancient times by explicitly taking into account soil–vegetation–climate feedbacks and spatial hydrological connectivity. Computations on a high-performance

computing system were sped up by parallelisation schemes that allow the application of this model to millennial timescales. The adjusted AquaCrop model was applied to a small catchment (11.4 km$^2$) in the mountain ranges of SW Turkey for the last 4000 years for which a detailed record on soil erosion and land use history exists. The model has first been calibrated against contemporary crop yield measurements in contrasting environmental settings in the region. The adjusted AquaCrop model allows exploration of the effects of changing climate, land use and soil thickness on crop yield. Important hydrological effects of Mediterranean soils such as re-infiltration and the infiltration excess mechanism were incorporated into the model. These allow better modelling of the water balance of soils at spatial and temporal resolutions that best match the spatial scale at which these processes take place, and thus allow for more realistic crop yield predictions. The model cannot only be used to simulate changes in crop yield under changing environmental conditions, it also allows evaluation of changes in catchment hydrology following soil degradation.

Within this study, we limited the application of the adjusted AquaCrop model to changes in climate and soil thickness following intense soil erosion, and this only for winter wheat. However, the adjusted AquaCrop model has the potential to simulate the effects of ancient land management techniques such as irrigation, crop rotation, fallowing, as well as manuring through changing the soil fertility stress indicators, on crop productivity. Archaeological and palaeobotanical data may help in defining realistic land management techniques as input to the crop yield model. An extension to more crop types is possible as long as local calibration against contemporary crop yields can be performed. As such, this modelling approach offers many opportunities in evaluating past human-environment interactions, including the resilience and adaptiveness of past societies to changing environmental conditions, which reaches far beyond traditional and theoretical debates on soil degradation through intense land use practices in the past.

**Author Contributions:** Conceptualization, M.V.L. and G.V.; methodology, M.V.L. and G.V.; software, M.V.L.; validation, M.V.L. and G.V.; writing—original draft preparation, M.V.L.; writing—review and editing, M.V.L. and G.V.; visualization, M.V.L.; supervision, G.V.; funding acquisition, G.V. All authors have read and agreed to the published version of the manuscript.

**Funding:** This research is funded by the Interuniversity Attraction Poles Program IAP 07/09, initiated by the Belgian Science Policy Office.

**Data Availability Statement:** Adapted AquaCrop code available at https://github.com/MaartenVL/aquacrop_spatiallyexplicit (available online from the 23 June 2021 onwards).

**Acknowledgments:** We thank Jeroen Poblome and other staff members of the Sagalassos Archaeological Research Project for their support and advice on the field campaigns. We are grateful for the help of Salih Ceylan and Hazal Elif Taşcı from Mehmet Akif Ersoy Universitesi and Oktay Darcan from Burdur Agricultural office, without whom field work would not have been successful. Ibrahim Erdal from Süleyman Demirel University is thanked for his support with the soil analysis. Finally, the computational resources and services used in this work were provided by the VSC (Flemish Supercomputer Center), funded by the Research Foundation—Flanders (FWO) and the Flemish Government—department EWI. We thank the reviewers for their constructive feedback. We also would like to honor Marc Waelkens who sadly passed away early 2021. Marc Waelkens was the founder of the Sagalassos Archaeological Research Project and its director from 1990 to 2013. He was a leading archaeologist in Classical Archaeology, and a pioneer of interdisciplinary archaeological research in Turkey. His holistic approach is being carried on through the many students he trained at Sagalassos. Not only will he be remembered as a renowned scientist, but also as a compassionate father-figure for the local community in Aglasun. He will be greatly missed.

**Conflicts of Interest:** The authors declare no conflict of interest.

## Appendix A

In the appendices the present day temperature and precipitation distribution map can be found. This is followed by a more detailed description of the field work, as well as

details on how the soil fertility stress parameter for AquaCrop was handled in this study. Finally, the generation of daily precipitation data using WeaGETS is explained.

*Appendix A.1. The Study Area of Sagalassos: Temperature and Precipitation*

There is a high spatial variability in climatic variables, due to the changes in topography. The NewLocClim [65] derived precipitation and temperature maps illustrate this in Figure A1.

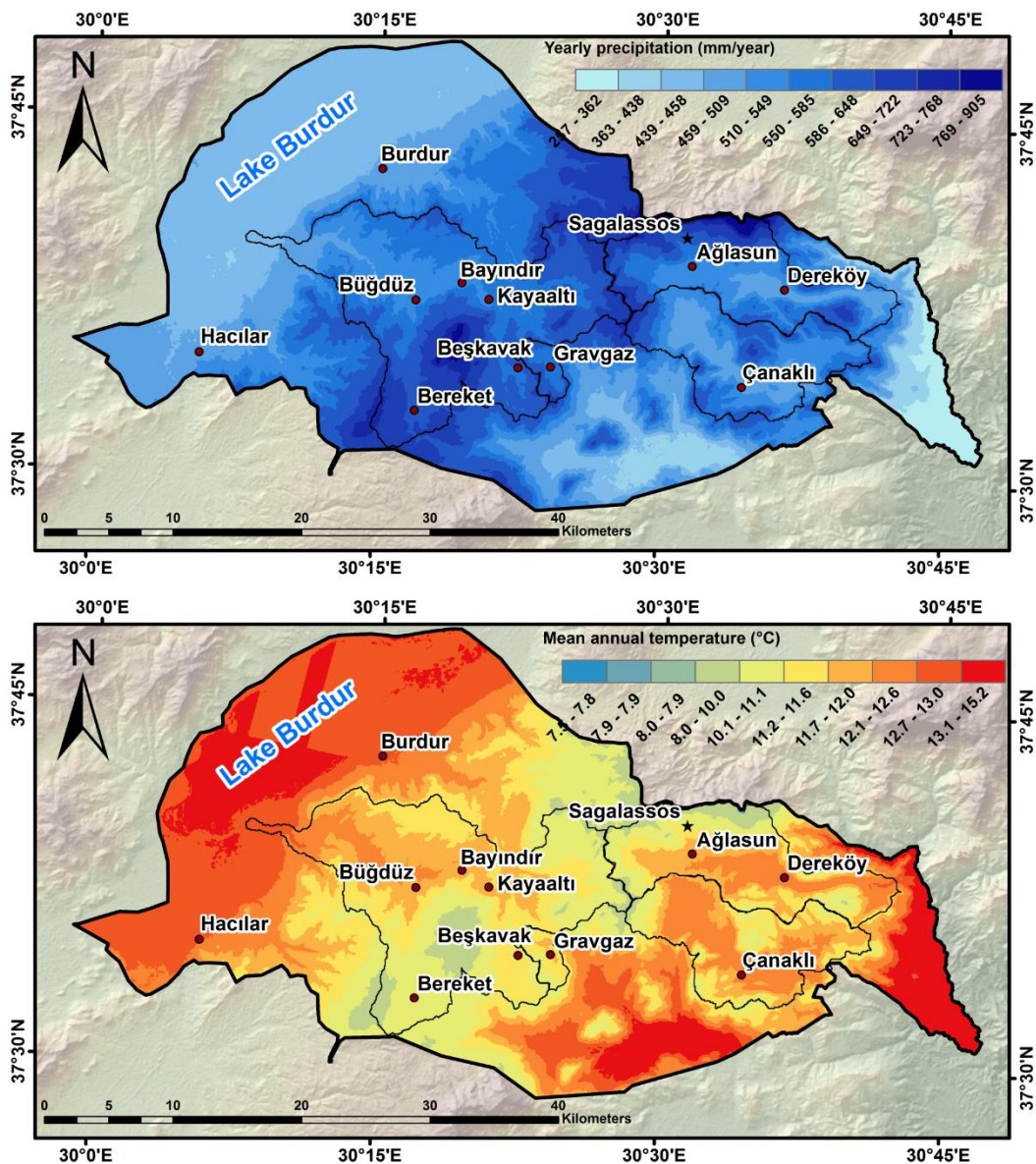

**Figure A1.** Yearly precipitation (mm/year) and mean annual temperature (°C) in the territory of Sagalassos, as derived from NewLocClim observational data [65].

*Appendix A.2. Calibration of the AquaCrop Model in the Environment of Present-Day Sagalassos*

Appendix A.2.1. Collecting Crop Yield and Plot Data: Field Work Details

The position of the plot was recorded with a Trimble GeoExplorer XT 2005 (2–3 m accuracy). An overview of all plot locations is provided in the appendices. Crops were cut at ground level from a 1 m$^2$ plot and total biomass was measured at the spot. A subsample was taken at every plot, weighed, and taken to the lab for further analysis. After having dried for at least 24 h in open air, the crop subsample was threshed, and the sub-sample's

crop yield (CY, t/ha/year) was measured. The crop yield of the plot was calculated by multiplying the subsample CY with the ratio of the sample mass to the subsample mass. The 38 measured CY data served to calibrate the AquaCrop model (Figure A2). Input data for the calibration runs were obtained either locally or using standard parameter values. For each plot, the stoniness was estimated using the grid method. A hand auger was used to estimate the depth to the bedrock and to take a soil sample between 15–35 cm depth. Initial Soil Water Content (SWC) could not be measured but was estimated to be low at the start of the growing season. A field work study by [66] determined the SWC in the study area to be between 15–25% in August. Initial SWC values at planting were assumed to be 15%. Local slope was recorded by means of a clinometer. For each plot, information on the timing of the different crop phenological phases of the past growing season and on local land management techniques was obtained through questionnaires with local farmers. Monthly temperature and precipitation data was provided by Burdur Meteorological Office. Daily precipitation was estimated using the monthly precipitation values from Burdur and the distribution of daily precipitation recorded at a nearby meteorological station in Isparta.

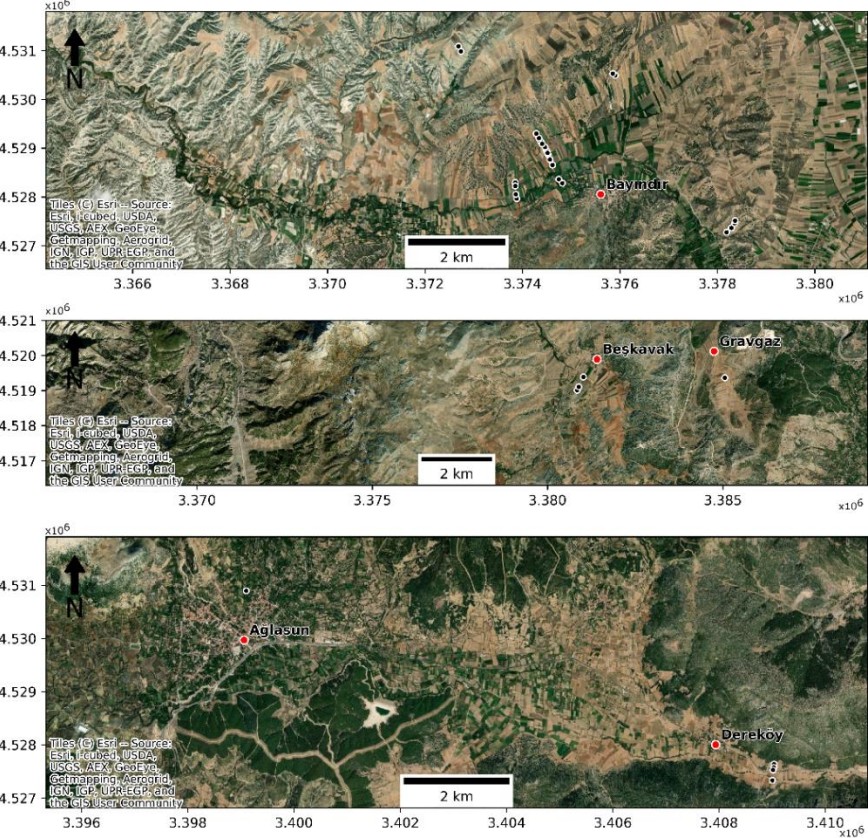

**Figure A2.** Locations of the 38 wheat plots for which crop yield and other plot and soil properties were measured in black. Top to bottom panels are indicated with respectively Grid's 1 to 3 on Figure 1.

Appendix A.2.2. Soil Fertility Stress

Calibration of AquaCrop is optimally done on fields without soil fertility stress [52]. A correct assessment of the soil fertility parameter is necessary to correctly calibrate the model. One way to map the stresses on a crop resulting from poor soil quality and nutrient deficiencies is to apply the semi-quantitative method as described by [67]. This approach requires a control field without soil fertility stresses. CY from this control field is compared with the CY on a field of choice with soil fertility stress, and the difference in CY is attributed to the soil fertility parameter. To overcome the lack of control fields in this study, we applied

an alternative method to account for soil fertility stress than proposed by [67]: a principal component analysis (PCA) of the measured plot and soil parameters was used to estimate soil fertility stress (Figure A3, Table A1). Principal components (PC) 1 and 3 are assumed to capture the effect of soil fertility on CY. High scores on both PC1 and 3 result in high CY (bottom-left panel), and the loadings from PC1 and 3 also suggest a soil fertility influence (Table A1). A soil fertility score was established for each plot by projecting the plot's scores of PC1 and 3 onto the line defined by PC3 = PC1 as shown in Figure A3, bottom-right panel. This score was then translated into the soil fertility parameter in the following way: a PCA derived soil fertility score larger than or equal to 2 was set to very light stress; a PCA derived soil fertility score larger than or equal to 0 and smaller than 2 was set to mild stress; a PCA derived soil fertility score larger than or equal to −1 and smaller than 0 was set to moderate stress; a PCA derived soil fertility score smaller than −1 was set to severe stress. The effects of the discussed stresses very light, mild, moderate and severe on CY are discussed in [31] and define the soil fertility parameter in AquaCrop.

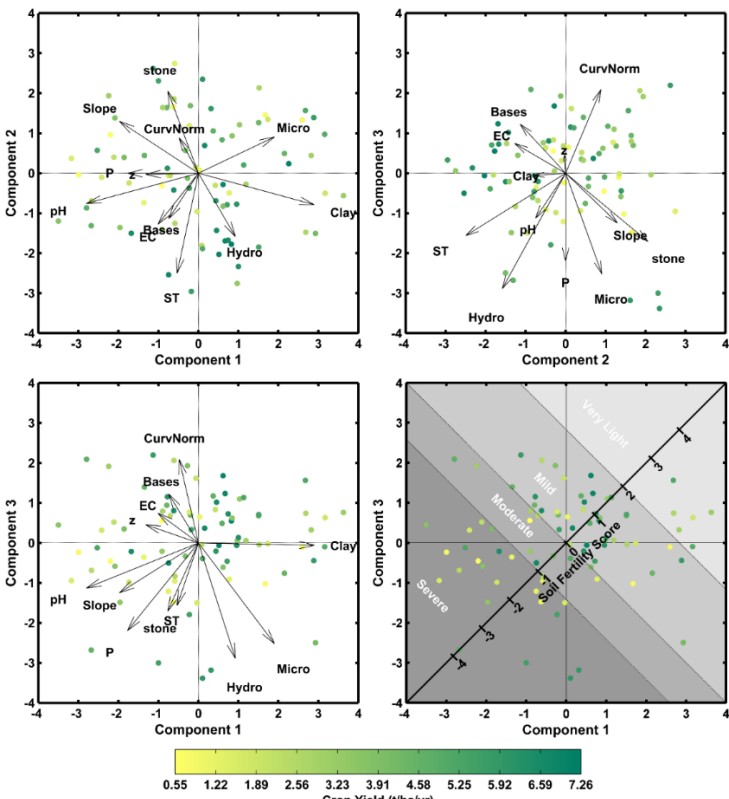

**Figure A3.** Top panels and bottom-left panel: Biplots of PC1 & PC2, PC1 & PC3 and PC2 & PC3. Bottom-right: assessment of the soil fertility parameter based on the PCA output: A soil fertility score was established for each plot by projecting the plot's scores of PC1 and 3 onto the line defined by PC3 = PC1: This score, was then translated into the soil fertility parameter in the following way: soil fertility score > 2: very light stress; 0 < soil fertility ≤ 2: mild stress; −1 < soil fertility score ≤ 0: moderate stress; soil fertility score ≤ −1: severe stress.

**Table A1.** PCA loadings for the first 4 principal components.

| Comp. 1—Explained Var: 0.23 | | Comp. 2—Explained Var: 0.15 | | Comp. 3—Explained Var: 0.13 | | Comp. 4—Explained Var: 0.11 | |
|---|---|---|---|---|---|---|---|
| Positive loading | Negative loading | Positive loading | Negative loading | Positive loading | Negative loading | Positive loading | Negative loading |
| Clay 0.37 | pH −0.52 | Stoniness 0.47 | ST −0.55 | Curvature 0.47 | Hydro −0.45 | z 0.59 | EC −0.49 |
| Micro 0.22 | Slope −0.39 | Slope 0.30 | Hydro −0.34 | Bases 0.31 | Micro −0.39 | Micro 0.36 | P −0.25 |
| Hydro 0.06 | P −0.36 | Micro 0.21 | EC −0.27 | EC 0.22 | P −0.33 | ST 0.24 | Hydro −0.24 |
| | z −0.29 | Curvature 0.21 | Bases −0.24 | z 0.17 | Stoniness −0.23 | Bases 0.15 | Stoniness −0.22 |
| | EC −0.24 | P 0.01 | Clay −0.17 | Clay 0.17 | ST −0.21 | pH 0.10 | Curvature −0.13 |
| | Stoniness −0.20 | z 0.00 | pH −0.16 | | Slope −0.15 | Slope 0.06 | Clay −0.05 |
| | Bases −0.20 | | | | pH −0.13 | | |
| | ST −0.16 | | | | | | |
| | Curvature −0.15 | | | | | | |

### Appendix A.2.3. Implementation of Trees and Shrubs in AquaCrop

Runoff was adapted trough the curve numbers consulted from USDA (1997) for typical native trees in the region, the Juniper cover type: 58, 58, 73 and for respectively HSG A, B, C and D. Transpiration and evaporation crop coefficients were estimated based on guidelines from [68]: crop parameter values for evapotranspiration (Kc) published in [69] were used. A value for Kc of 0.95 is listed, which is lower than the value for winter wheat (1.1). Maximum rooting depth was estimated to be 3 m, based on values from [70].

### *Appendix A.3. Paleoclimate Time Series*

Downscaling was done by comparing our data with the CRU TS 1.2 dataset [71], which combines monthly climate observations for roughly the last century on a resolution of 10′ by 10′ in Europe. The ECBilt-CLIO-VECODE output data for the last century was resampled to the 10′ by 10′ resolution and the deviations from the CRU TS 1.2 dataset were calculated and applied to the resampled ECBilt-CLIO-VECODE output. We assume that this downscaling factor remained the same the past 4000 years, and that the downscaled climate data within the 10′ by 10′ grid cell also applies to Gravgaz, ignoring further topographic effects on precipitation at an even finer scale within the 10′ by 10′ grid cell. Details on the downscaling approach can be found in [72,73].

Figure 7 shows the yearly temperature and precipitation time series for the last 4000 years, as well as the PDF of temperature and precipitation for the different archaeological periods. Both temperature and precipitation show a decline towards the present, although there is very few temporal variation in both temperature and precipitation time series.

AquaCrop, however, requires daily precipitation data. To get a higher temporal resolution from the palaeoclimatical timeseries, the WeaGETS stochastic weather generator was used in combination with observed daily precipitation records in the territory.

Contemporary daily precipitation data necessary to downscale the monthly precipitation time series were retrieved from the European Climate Assessment and Dataset (ECA&D) [74]. This was done for the nearby meteorological station in Isparta (37.75 N, 30.55 E, 997 m.a.s.l). ECA&D collect daily meteorological observations for stations across Europe. A time series was obtained for Isparta for 1931–2008, however, over 50% of the days lack precipitation data.

WeaGETS uses a two-parameter gamma distribution to generate daily precipitation, given by Equation (A1). The gamma distribution is widely used to model precipitation amounts [75]. It can capture a whole range of distribution types (from exponential decay to normal distributions) and is hence applicable in many situations [76]:

$$f(x) = \frac{\left(\frac{x}{\beta}\right)^{\alpha-1}\exp\left(-\frac{x}{\beta}\right)}{\beta\Gamma(\alpha)} \tag{A1}$$

with $\alpha$ and $\beta$ the two distribution parameters, $x$ the daily precipitation and $\Gamma(\alpha)$ the gamma function evaluated at $\alpha$. The mean of the gamma function is given by $\alpha*\beta$. $\alpha*\beta$ values were derived for each year from the palaeo-climatical precipitation series. To simulate

daily palaeoprecipitation values, the WeaGETS model was run with a gamma function established from the present-day daily precipitation record from Isparta, where the mean of the gamma function was changed according to the difference between present-day $\alpha$ and the $\alpha$ derived from the palaeoprecipitation time series.

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
