# Peer review of "A Spatially Explicit Crop Yield Model to Simulate Agricultural Productivity for Past Societies under Changing Environmental Conditions"

_water, doi:10.3390/w13152023_

Round 1

Reviewer 1 Report

The paper is well structured. The language of the manuscript is perfect. The methodology is sound and the results are corroborating the assumptions.

It is impressive to be able to run the agronomic AquaCrop model for the last 4000 years. It is definitely of great scientific interest to simulate the impact of climate and land cover changes, as well as soil dynamics, on the productivity of winter wheat crops for a Mediterranean mountain environment such as the south west of Turkey. However, I am afraid that despite the fact that such climate-based models can be very useful to create informed and realistic models of obtainable crop yields in the past, but agricultural management practices are known to increase yields substantially, and those practices are not factored into environmentally-derived models and also some social factors could be ignored or not well taken into consideration, without mentioning the seasons of pests for example that were not considered but as a modelling work it was well dealt in this work.

I suggest to the authors to check for example the works of Currie et al such as: “Agricultural productivity in past societies: toward an empirically informed model for testing cultural evolutionary hypotheses”, and at least cite them in the paper introduction. That said, I saw that the authors already cited Oyebamiji et al “Emulating global climate change impacts on crop yields”, so it is ok.

The calibration and validation of the model have been properly carried out. Some assumptions made to ease the calculation using Matlab could be questioned, but in general it makes sense.

I was expecting to be honest greater results, but it is still interesting and it is definitely a promising research area. It is going to help us a lot on the way to discover the adaptation and the resilience of ancient civilizations and probably get to know the reasons of many historical facts.

I encourage the authors to proceed further in their works and publish the results of their simulations using the AquaCrop model in particular in what is related to land management.

Author Response

I sometimes refer to the revised manuscript, I can't upload 2 documents here, so I will ask the editor to provide you with this as well.

Reviewer 2 Report

The objective of this paper was to investigate the sustainability of wheat yields in an area of Turkey with an agricultural history of about 4,000 years using a crop simulation model. The project focused on yield response to rainfall and temperature as affected by soil erosion and deforestation over this period. The paper is well written and easy to read.

There are a lot of difficulties in a project like this that goes far back in the past where there is little recorded agricultural history. The authors addressed many of the issues well. I can understand that the authors chose to consider only a modern variety for all the simulations as there may not be enough information on more ancient varieties to simulate those varieties’ responses to the environment. So, the yield responses are largely relative and the authors did not read too much into the yields themselves. However, the assumption that runoff and soil erosion would be the same under varieties and cultivation practices used in ancient times as they are now is not  realistic and the authors should address this. There were differences in plant density and plant cover over the years due different planting methods, seed density, availability of fertilizer, etc. Planting in rows was not a common practice in the past – it is likely there would be sparse plant density in some locations – how would this affect runoff? I would think there would be a lot more variability with older practices.

But, I think the important message from the paper would be that soil degradation has reduced potential yield relative to what could have been produced had there been no degradation.

Overall, I think research like this is helpful and I think the paper is publishable with minor modifications.  My specific comments follow

In the abstract “changing environment” is commonly interpreted as pertaining to weather/climate only. I think the authors here mean geophysical and climatic environment?

Line 112, I suggest changing ‘own’ to ‘local’ or ‘on-farm’

Line 132 ‘Recent years’ – it would help to tie a period to this -last 100 years/ 10 years?

Line 135 Sediment transport does not leave the lake? What about accumulation of sediment in the lake and the effect on water level?

Line 152 – change ‘is’ to ‘are’ as in data are…

Line 153 – I would recommend  ‘complex’ rather than ‘demanding’  After all, a model that crashes all the time can be considered ‘demanding’…

Line 154 change ‘come to’ to ‘determine’ or ‘estimate’

Line 167 How is the change in water holding capacity addressed?

Line 173 figure caption, drop ‘actually’, there is no need for a modifier. The water either can infiltrate or it cannot.

Lines 186 to 193 – is ponding in depressions accounted for?

Lines 242 to 245. I can understand why the authors need to do this. But, it is a significant assumption that the use of modern management practices (row spacing, fertilization, tillage, etc) and variety (length of growing season, progress of crop cover, etc) to simulate ancient agriculture will not have a big effect on runoff or erosion. Many would not agree with this assumption. It would be important for the authors to address this.

Line 305 trough -> through

Line 346-352 were there effects of sediment accumulation on the water height? If there was erosion, where did the soil go?

Line 362 - is 0.3m2 the size of the drainage opening? Is this correct?

Line 369 – assume water temperature is the same as the air?

Figure 8 – was the range of weather variability in the calibration data set similar to that in the simulations?

Table 2 Tbase seems very high, can the authors comment on this? How does it affect the calibration results? it seems that maybe there were not enough calibration data with very cool temperatures. 

Table 2. Please define WP and CHC here in the table. It seems these variables would be most unfamiliar to people who have not used the model.

Lines 412-416. I would rewrite this or delete. To me, it reads like “we had no data to check the model and since we have no data for previous time periods, the model will work just as well as for the cases where there were no data.” In my opinion, it is a poor argument for not having data for the past.

 There is also a double negative in this statement “lack of data… is often also not available…” à and it actually means there were data available.

Line 423 – the authors are really talking about a relative yield correct? I think not saying otherwise would give readers the idea that these were the yields farmers in those days observed.

Figure 11, please define LU and ST in the figure caption (makes it a bit easier for the reader).

Author Response

(The authors gave the same response as above.)
